# Barriers and perceptions of WHONET/BacLink adoption in Nepal: A qualitative study of clinical microbiology laboratories

Sanju Maharjan[1]*, Patrick Gallagher[2], Manish Gautam[1,3], Sanjay Gautam[3], Manisha Budhathoki[1], Reena Mukhiya[4], Smirti Kattel[5], Amit Bhandari[6], Hea Sun Joh[3], Ahmed Taha Aboushady[7], Raphaël M. Zellweger[3], Madan Kumar Upadhyaya[8], Runa Jha[9], Jyoti Acharya[9], William R. MacWright[2], Florian Marks[2,10,11,12], John Stelling[6], Nimesh Poudyal[3]

1 Anweshan Private Limited, Lalitpur, Nepal, 2 Public Health Surveillance Group LLC, New Jersey, United States of America, 3 International Vaccine Institute, SNU Research Park, Gwanak-gu, Seoul, Korea, Republic of Korea, 4 Burnet Institute, Melbourne, Australia, 5 University of the West of Scotland, London Campus, London, 6 Independent Health Expert, Kathmandu, Nepal, 7 Brigham and Women's Hospital, Harvard Medical School, Boston, Massachusetts, United States of America, 8 Quality Standard and Regulation Division, Ministry of Health and Population, Government of Nepal, Kathmandu, Nepal, 9 National Public Health Laboratory, Department of Health Services, Ministry of Health and Population, Kathmandu, Nepal, 10 Cambridge Institute of Therapeutic Immunology and Infectious Disease, University of Cambridge School of Clinical Medicine, Cambridge Biomedical Campus, Cambridge, United Kingdom, 11 Heidelberg Institute of Global Health, University of Heidelberg, Im Neuenheimer Feld, Heidelberg, Germany, 12 Madagascar Institute for Vaccine Research, University of Antananarivo, Antananarivo, Madagascar

* sanju@anweshan.org

## Abstract

### Background

The International Vaccine Institute-led CAPTURA (Capturing Data on Antimicrobial Resistance Patterns and Trends in Use in Regions of Asia) project delivered capacity building activities to strengthen antimicrobial resistance surveillance activities in Nepal.

### Methods

The CAPTURA project trained 97 laboratory personnel from 19 hospitals on the use of WHONET/BacLink software to manage microbiology data in Nepal during 2020–2021. Approximately two years later, the trainees were followed up by phone to assess implementation status and effectiveness of the training. An inductive approach was used for coding and categorization of their response, and themes were generated for analysis. Trainees from ten hospitals agreed to respond regarding their experience.

### Results

We found that two out of the ten hospitals were using the WHONET/BacLink software, with one each within and outside the national AMR surveillance network. The

**Data availability statement:** The data supporting the findings of this study are available in the supplementary table (S4 Table).

**Funding:** The "Capturing Data on Antimicrobial Resistance Patterns and Trends in Use in Regions of Asia (CAPTURA)," project at the

International Vaccine Institute was funded by the Department of Health and Social Care's (DHSC) Fleming Fund using UK aid. Grant number IVI FF10/135. The funders had no role in study design, data collection and analysis, decision to publish, or preparation of the manuscript.

**Competing interests:** The authors have declared that no competing interests exist.

remaining eight hospitals never implemented the system despite receiving the training. Key barriers to implementation included, hospital administration prioritizing other interoperable software, limited ongoing training, inability to export data from an LIS, limited real-time assistance with technical issues, and poor confidence in analyzing data. In addition, limited human resources and minimal capacity-building activities resulted in a lack of confidence in using the system independently, which were also identified as barriers.

## Conclusion

Implementing WHONET/BacLink software in hospital settings can be challenging due to various factors, including a lack of knowledge and confidence among users, a lack of time and human resources to use the software effectively, and a lack of interoperability with other hospital management systems. Real-time support and follow-up activities potentially reinforce the skills and knowledge delivered during the training.

---

## Background

Effective detection, prevention, and response to antimicrobial resistance (AMR) threats require a collaborative approach, such as establishing coordinated national and international surveillance systems [1]. Global Antimicrobial Resistance and Use Surveillance System (GLASS) was launched in 2015 as a strategy to respond to gaps in AMR surveillance and globally integrate the existing systems to strengthen evidence-based AMR surveillance. One of the systems to manage clinical microbiology data and participate in the GLASS platform [2] is publicly available software. The application WHONET is designed to study the epidemiology of infectious agents, guide the rational use of antimicrobial agents for laboratory testing, detect hospital and community outbreaks, and identify issues with laboratory performance [3].

AMR is a global public health priority that requires a uniform data reporting surveillance system. Meanwhile, the WHO's Global Antimicrobial Resistance and Use Surveillance System (GLASS) incorporates a 'Tricycle' and 'Transdisciplinary one health approach' that recognizes the interconnection between human, animal, and environmental health through built-in harmonized WHONET application including BacLink to facilitate automatic scheduled uploading of data from local computers [4,5]. The WHONET/BacLink software allows microbiology laboratory data conversion to a uniform format and has been proven to be an effective alternative to multiple platforms tested and practiced [6]. The CAPTURA (Capturing Data on Antimicrobial Resistance Patterns and Trends in Use in Regions of Asia) project provided training to use the WHONET/BacLink to laboratory staff of various sites/hospitals in Nepal. Following this training and intervention, hospitals were expected to adopt WHONET/BacLink for laboratory data management. However, the extent of its implementation varied. This paper explores the perceptions of laboratory staff regarding the adoption of WHONET/BacLink and examines the barriers and challenges they encountered in integrating the software into their hospital systems.

## Methods

Representing 28 CAPTURA partner sites (11 AMR surveillance sites and 17 outside the national AMR surveillance network), 97 trainees, including laboratory personnel and health workers of 19 hospitals, received two-day training during 2020 and 2021. Two trainers from the International Vaccine Institute (IVI), who were remotely trained through CAPTURA project provided the training. From the participants' information collected during the training, a follow-up call was set up at least 12 months after the training to receive feedback on software usage. A qualitative phone interview was conducted with the trainees to understand the effectiveness of training on WHONET/BacLink and to provide policymakers with insights into the obstacles that hinder the sustainable adoption of these methods for antimicrobial resistance (AMR) surveillance systems. The training was provided exclusively to laboratory personnels expressing their interest in the WHONET/BacLink system. The training conducted in native language (Nepali) covered topics such as an introduction to the software, laboratory configurations, data extraction, data cleaning, and data analysis using the WHONET and BacLink systems. Personal laptops and computers were used for practical sessions during the training. The question-and-answer session was taken after the completion of the interactive training session each day.

From 15th July 2022 to 8th August 2022, qualitative interviews were conducted based on semi-structured interview guidelines. The enlisted trainees were asked 15 questions (S2 Table) to understand the current situation of system applicability. Verbatim translations were performed for all recorded interviews. The recordings were stored in a password protected drive with access limited to four team members who were involved in coding and data review process. The translations were anonymized, and all the recordings were destroyed after transcription and translation of the interviews. The translated raw data was reviewed multiple times for coding purposes, and new codes were generated using an inductive coding method. The codes were iteratively refined for consistency and experts including country lead for CAPTURA project also reviewed the generated codes, contributing to the validation process and enhance the reliability. NVIVO (https://libguides.library.kent.edu/statconsulting/NVivo) and Microsoft Excel were used for coding and data management [7]. A conceptual model for capturing implementation barriers and challenges to adopting WHONET/ BacLink has been presented in Fig 1.

The research proposal and data collection tool were approved by the internal research team at Anweshan in the guidance of CAPTURA team from International Vaccine Institute (IVI). This is a program evaluation assessing the effectiveness of capacity building activity, and therefore it is not classified as human subject research. Each participant provided a verbal informed consent for their participation during the follow up and was approved by IRB on the process of documentation. The consent was documented in the audio recording and witnessed by a fellow researcher while interviewer was taking an interview.

## Results

Out of 19 hospitals (including ten privately owned, seven publicly owned, and two NGO-based hospitals) where WHONET/BacLink training was conducted, ten hospitals (Five from within the AMR surveillance network and five from outside) participated in the interviews for the assessing the experience barrier and challenges while implementation (Fig 2). This included three private hospitals, six public hospitals, and one NGO-based hospital. In nine hospitals (out of 19) where training was implemented, trainees could not be enrolled in an interview for this assessment mostly due to relocation and transfer to other hospitals or sites where there is no application of WHONET/BacLink system.

We found that two out of 10 hospitals (one from within the network and the other one from outside) were using the WHONET/BacLink as an ancillary software for recording and reporting AMR data. This indicated that it served as an adjunctive tool rather than primary system at hospital. These two hospitals using WHONET recorded variables such as the origin, age, sex, location, institution, department, specimen date, organism, disk diffusion method, and specimen type in the system. However, only one hospital recorded other variables such as specimen number, details on screening for important drug resistance markers of special interests such as: extended-spectrum beta-lactamase (ESBL) and

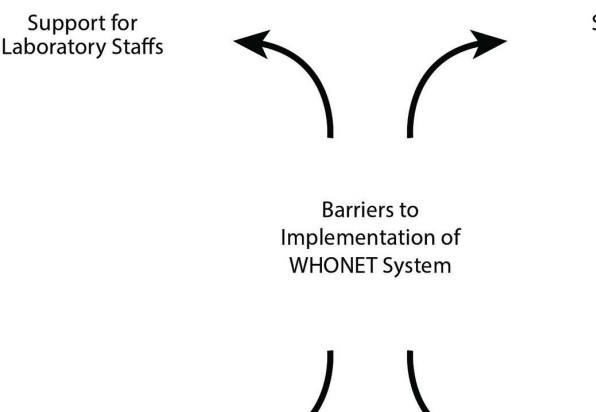

- Lack of technical support and guidance
- Literacy and skill in software
- No training or refresher training

- Less involvement of hospital management
- Poor system integration and interoperability
- Poor capacity building approach
- Resources constraints

Support for Laboratory Staffs

System within Hospital

Barriers to Implementation of WHONET System

WHONET and Backlink System in Hospital

Data Use and Analysis

- Impact of WHONET training
- Lack of users involvement
- Problem reaching to patient's end
- System usability and practice

- Less practice of data analysis and reporting
- Less knowledge on system usability

**Fig 1. A model representing barriers to WHONET/BacLink implementation in Nepal.**

methicillin-resistant *Staphylococcus aureus* (MRSA), and location type. In terms of specimen type, blood, urine, pus, and body fluids were recorded by both sites, while information on sputum, stool, tissue, and catheter specimens was available in one site. The percentage Resistance, Intermediate and Susceptibility (RIS) were calculated in both hospitals, while isolate listing and quick analysis was practiced in only one site.

Despite being offered training, six out of ten hospitals did not utilize the WHONET/BacLink software and showed reluctance to adopt it in the foreseeable future. Among the four hospitals that initially implemented the software after training, only two continued to utilize it up to the follow-up date. The explanations behind not using the WHONET/BacLink system by these hospitals were collected, coded, and categorized as themes and sub-themes for further analysis. Some of these include an interoperability of existing systems, a lack of technical support, and experienced gaps in data usage and analysis.

## Effectiveness of training on WHONET/BacLink

Participants provided predominantly practical feedback on the training, expressing that it was beneficial for recording, reporting, and analyzing AMR data. They also expressed appreciation for the training's content and methods. Some participants mentioned that the demonstration using dummy data from hospitals was impactful and helped them to understand the potential issues with recording and reporting data. However, some participants felt that the training duration was

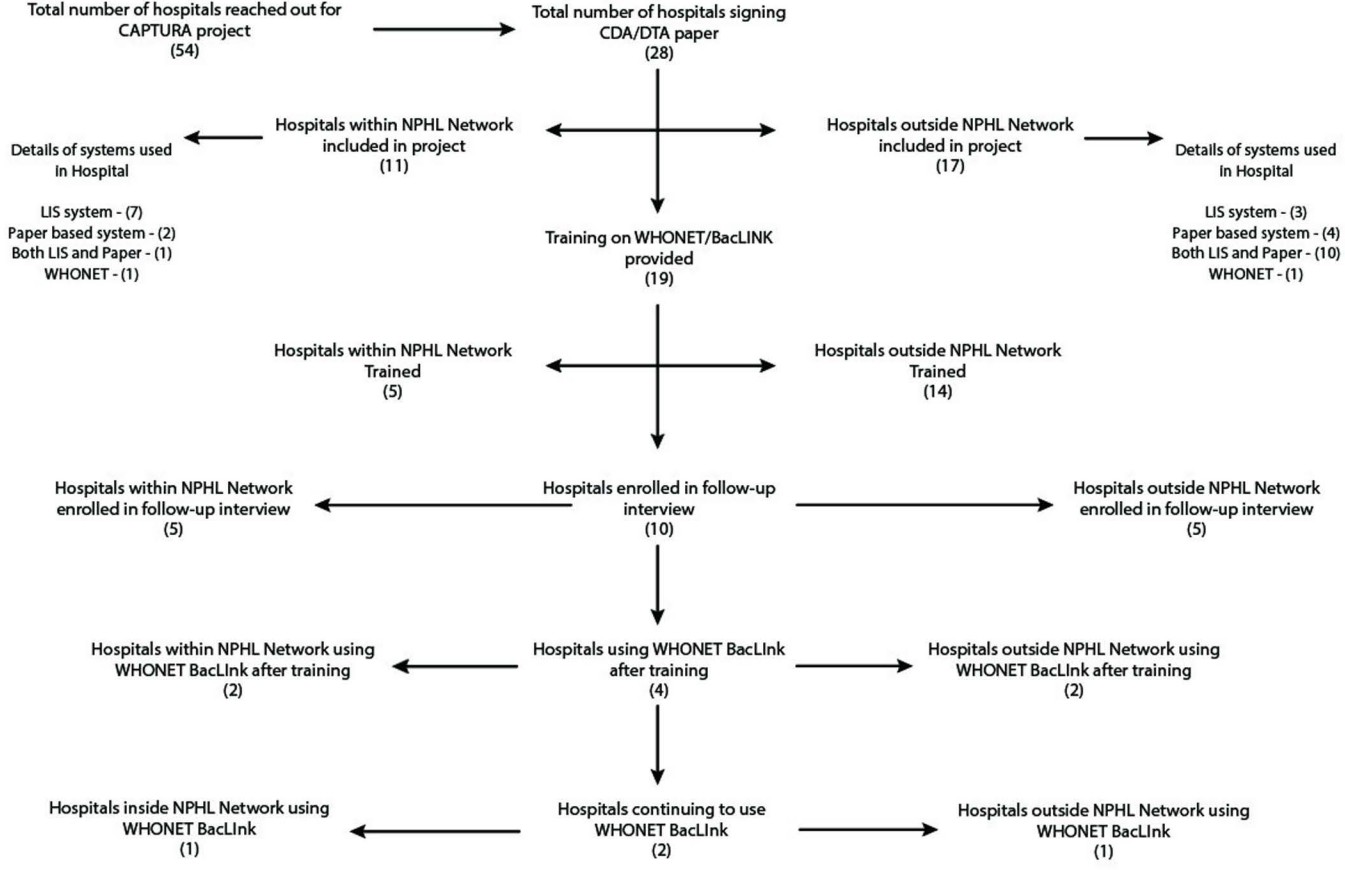

**Fig 2. Total number of hospitals enrolled in this follow-up interview.**

insufficient though on-site real-time practice on data was provided. A few participants suggested that the training would have been more effective if a support team had been deployed to each hospital for at least a week following the training.

One of the key activities undertaken during the CAPTURA project period was capacitating the hospitals on exemplifying the importance on use and sharing of data for decision making practices through the use of WHONET/BacLink system as a software at project sites. The designated staff at the hospital were provided training during the period. The analyzed information shows that a small number of hospitals (four out of 19) implemented the WHONET/BacLink system after receiving training (Fig 2). Two of these continued to use the system adjoinedly but effectively for recording, reporting, and disseminating reports. The other two hospitals within the national AMR surveillance network switched to different software after the hospital authority introduced a new system. However, one hospital within the AMR surveillance Network has been using the WHONET system since 2014 and reported frequent use of the software for patient reporting and analysis following their introduction to BacLink in 2020.

The hospitals using WHONET/BacLink reported extensively using the printing function of the system as it eased the reporting process. The printed analysis reports were sent to doctors for decision-making on drug prescriptions. Though WHONET has the feature of modifying the clinical report in print option, it seemed like the participants were unaware about the available basic feature of WHONET including printing function. Because the major problem reported by the participants during an interview was that the system did not allow for the customization of the printing details for patients.

As a result, most users could not add or remove information as needed, making the printed reports incomprehensible to patients. Some of the trainees who underwent training suggested the development of a hospital specific standardized format/template for generating reports, with the intention of sending them on a monthly basis to the national reference laboratory (NRL) specifically designated for housing national data.

**Barriers to implementation of the WHONET/BacLink system**

We identified three key obstacles to implementing the WHONET/BacLink system in hospitals in Nepal:

i) **Interoperability of existing system**

The hospitals where training was provided were already using some types of electronic medical record (EMR) software to capture patient data, laboratory and radiological findings and generate and communicate reports. In CAPTURA sites, none of the hospitals within or outside the national AMR surveillance network exclusively used the WHONET system for AMR data recording and reporting. In settings where WHONET was not routinely used, AMR data is extracted in MS Excel format for internal reporting and shared with national data center when required. However, these hospitals could share patients' complete profiles, including laboratory reports and billing details, within their departments using existing EMRs. The main challenge stemmed from integrating WHONET data with existing EMR systems, which handle comprehensive patient profiles and billing details. Key issue was limited engagement from LIS developers and vendors in enhancing data cleanliness, particularly in minimizing the presence of free text and disorganized data structure. This issue is compounded by deficient exports from LIS systems, requiring manual cleaning and continual maintenance efforts. In addition, the incompatible file format of WHONET and EMR system added additional work in data management. Despite the analytical capabilities (such as quick analysis, isolate listing, and percent RIS) of WHONET being well-regarded, its implementation was hindered by resource constraints and the need for seamless interoperability with hospital-wide systems to avoid duplication of work. Introducing new software is challenging, particularly in settings with limited human resources. From the management perspective, refresher training to the staffs, retaining trained staff to perform the task is equally difficult, as operating the WHONET/BacLink is considered optional in most of the CAPTURA sites, requiring extra time and indirect cost investment. We observed that frequent shifts to new systems within the hospital setting or even the use of multiple systems, combined with inadequate technical support, contributed to a lack of motivation in ensuring the recording of high-quality and comprehensive data. This issue is particularly pronounced in resource-limited settings with high burdens, where the constant changes in systems can impede the commitment to data recording. Though BacLink can import data from diverse systems to WHONET compatible datasets that can be further analyzed and encrypted, the fragmented system with multiple software functional within hospital is what hinders the adoption of WHONET/BacLink completely. It is important that all departments and units within the hospital system are oriented on applicability of WHONET/BacLink before its adoption. This orientation should address not only technical issues but also ensure interoperability and the optimal utilization of available resources within a hospital setting.

ii) **Lack of adequate technical support and knowledge retention**

During WHONET/BacLink follow up assessment of the training, participants reported that the training on WHONET/BacLink provided through the CAPTURA project was the latest training conducted for laboratory staff on AMR data management and quality reporting. However, as reported, not all relevant staff from each hospital participated in the training, and some participants were sent on an ad hoc basis as trainees by the hospital administration. This highlights the challenges associated with training the relevant staff members and retaining knowledge within the departments where expertise is needed; for example, many of the trained staff quit their jobs or were transferred to a different unit within the laboratory. Similarly, staff turnover at most hospitals reportedly impacted the effectiveness and outcome of the training. Participants reported that the successful implementation of WHONET/BacLink in a particular hospital largely depended

on the staff who received the training. However, transferring knowledge to new staff members was challenging due to the unavailability of the codebook, user's handbook, or guidebook prepared after the training by hospitals with customization to fit their needs and usability better.

A recommendation was made to allocate a skilled professional to each facility to provide regular support and maintenance for the software. Participants emphasized the importance of timely response and the presence of support personnel to instill confidence in using the WHONET/BacLink system. They stressed the significance of continuous technical support for any potential issues while using the software. This aspect was seen as critical to the success of the surveillance system and the establishment of WHONET as the primary laboratory management system. Therefore, having a dedicated and trained support team available is crucial to ensure smooth operation and widespread adoption of the software by hospitals.

Finally, participants reported needing refresher and extensive training on WHONET/BacLink. Ongoing capacity-building activities were minimal as many participants left their jobs or failed to pass on the acquired skills and knowledge to the delegates. Although the training session included a demonstration of data analysis and data entry from real-time extracted data from the hospital system itself, loss of practice among participants on data and use of the system after completion of training resulted in recall bias. More than half of the participants had less confidence in using BacLink compared to WHONET due to a limited ongoing training and capacity building activities.

### iii) Discontinuity in data use and analysis

The WHONET/BacLink system was primarily used for reporting and internal record-keeping for microbiological data in hospitals networked with national data center for AMR surveillance. The National Public Health Laboratory (NPHL) in Nepal reports to GLASS using WHONET. All laboratories as part of AMR surveillance network are expected to share data with the NPHL using WHONET and send SQLite files for upload. Only one of the hospitals using WHONET is found sending SQLite file while reporting to NPHL whereas in other hospital, although the trainees used the WHONET system to update their monthly reports, we found that the records were initially sent as MS- Excel files rather than SQLite files due to a lack of confidence in using BacLink. Participants lacked awareness and proficiency in utilizing the full analytical capabilities of WHONET beyond basic functions such as percent RIS mapping and quick analysis. Many participants were unaware of the specific analysis methods that can be adopted in the WHONET system. Despite prior understanding, trainees struggled to recall and effectively utilize analytical methods within WHONET due to infrequent use and practice. A critical shortcoming was identified in the absence of alert settings for antibiotic breakpoints within the system, indicating a potential oversight in monitoring and responding to emerging AMR trends. Furthermore, trainees exhibited inconsistencies in interpreting AMR trends, resistance profiles, cluster alerts, isolate alerts, and scatter plots within the software, highlighting a need for enhanced training and guidance.

Overall, participants suggested that a periodic basic to advanced level WHONET/BacLink course coupled with continuous refresher's training would be necessary for sustainable adaptation and improvement of WHONET/BacLink system. In addition, having the capability to upload complete patient records into WHONET/BacLink and integrate them with other features of existing EMRs would facilitate the generation of more comprehensive analytical reports for hospital usage. However, regular use of WHONET in hospitals is challenging mainly due to the need for customization and improvement in various areas (S1 Table).

## Discussion

With a primary objective to identify the WHONET/BacLink implementation barriers in CAPTURA project sites in Nepal, we report key challenges and mitigation measures to make sustainable use of AMR surveillance systems in a resource-limited high-burden setting. In addition, this assessment focuses on measuring the impact of training to use WHONET/BacLink.

Introducing a new digital surveillance system is challenging, and the outcome depends on multiple factors, such as the usefulness and easy functionality of the system, the availability of well-acquainted human resources and the available technical support system [8]. Therefore, the implementation process undergoes multiple stages before its optimum utilization. The technology acceptance model (TAM 1989) suggests that when users are presented with new technology, perceived usefulness and ease of its use influence their decision on how and when they will adopt or use the system. User attitude, behavioral intention, and actual use of technology were added as additional external factors into the extended TAM model 2018 to explain factors affecting the acceptance of new technology in a particular setting [8,9]. This model has implications for the acceptance and use of the WHONET/BacLink system in hospitals. How users perceive the software, and the results of their training determine the implementation status. The findings suggest orientation to the entire LIS team, laboratory personnels and hospital management authorities to have optimum applicability of the software across the hospital and laboratories.

Interviews with trainees revealed a limited understanding of WHONET/BacLink's built-in features and its restricted functionality within hospitals participating in the national AMR surveillance network. Specifically, its use for distributing clinical reports and managing internal records was not well integrated. Instead of utilizing the standard SQLite format required for BacLink, hospitals extracted data in MS Excel files, indicating gaps in data management practices. Additionally, there was no clear plan for data analysis or utilization using WHONET/BacLink at the interviewed sites. These findings highlight the critical need for a structured approach to data collection, analysis, and reporting, ensuring alignment with a standardized framework to enhance the system's effectiveness across hospitals.

The lack of awareness among many participants regarding WHONET/BacLink's analytical functions underscores the need for more comprehensive training and continuous support. Strengthening users' familiarity and confidence with the system is essential to maximizing its potential. Additionally, frequent staff turnover in hospitals has likely contributed to resource gaps, hindering the effective implementation and sustained use of the software. While only a small number of expert users are required per site, the high turnover rate highlights the importance of training all relevant staff and ensuring continuous support through refresher training.

To address these challenges, a structured **Training of Trainers (ToT) program** should be implemented to create a pool of skilled trainers capable of providing technical guidance and refresher sessions as needed. At the national level, a centralized network of WHONET/BacLink trainers should be established to serve as a dedicated resource for hospitals. This would ensure the availability of expert support, facilitate the retention and utilization of trained personnel, and promote the long-term sustainability of WHONET/BacLink adoption across healthcare facilities.

The Delone and McLean (D&M) model suggests that a system's success depends on several factors, including system quality, information quality, user satisfaction, and net benefits [10]. In the case of WHONET/BacLink, it was essential that the system provided high-quality data records and reporting and offered users a high level of satisfaction and tangible benefits. Without these elements, it may be difficult to implement the system successfully at all hospitals. In this context, users showed a strong preference for EMR systems with interoperability across hospital departments, as they streamlined workflows and reduced redundant reporting tasks. This finding aligns with insights from the supplementary article, "Takeaways from the CAPTURA Project," which emphasizes how fragmented systems—characterized by multiple software platforms with overlapping functionalities—slow down processes and discourage users from adopting new systems for data management and reporting [11].

The WHONET/BacLink system is only for microbiology laboratories, thus cannot be used in other departments. The users perceived WHONET as effective from their perspective; nevertheless, at the institutional level, implementing WHONET required them to put in additional hours to input the required information for WHONET given that the system lacked interoperability feature. This overshadowed the benefits of WHONET, suggesting improvements in a system such that WHONET/BacLink meets the minimum expectation. To enhance WHONET/BacLink adoption, a comprehensive assessment of the hospital's digital ecosystem should be conducted before implementation to ensure seamless integration

with existing systems. Additionally, national-level policies should prioritize system integration strategies, and a dedicated task force or technical support unit should be established to assist hospitals in optimizing WHONET/BacLink within their workflow.

We identified multi-faceted barriers to successfully implementing a new system. Enhancing the capacity of laboratory staff by allowing them to use the WHONET/BacLink system regularly would build confidence towards adopting the approach [12]. However, this process must remain continuous and adaptable to evolving needs. This follow-up study highlights key challenges and possible solutions in integrating the system into hospital settings (S3 Table). With these insights, hospital authorities and national stakeholders can collaborate to address the identified barriers and develop effective mitigation strategies. Importantly, we appreciate that hospitals have no "One size fits all" model to roll over the WHONET/BacLink system. Whether hospitals are within or outside the national AMR surveillance network, including laboratory personnel in the decision-making process was crucial for AMR data management. Though hospital administration is critical to introducing any system, successful implementation depends primarily on users' motivation and perception.

Our study has several limitations. First, we focused on tracking previously trained individuals and assessing the barriers they encountered in adopting WHONET/BacLink. As a result, the sample size is relatively small, limiting the generalizability of our findings to other hospitals. Also, this study does not compare hospitals adopting the system with the ones that did not due to fewer sample included for this follow up study. Additionally, future research should consider including higher-level laboratory management, because during this intervention training was focused more to laboratory staffs only. While these officials approved the training and deployment of WHONET, factors such as system interoperability, staff retention, and the discontinuation of system use due to the lack of refresher training emerged as challenges during implementation. Staff turnover also affected our ability to assess all 19 hospitals initially included in the project, restricting this study to phone interviews with only 10 hospitals. To minimize response and recall biases, we designed the survey questions with specific reference to the project intervention period. Furthermore, the interviewer, who was involved throughout the project, conducted the data collection to ensure consistency. Interviewer bias was also mitigated by avoiding verbal and non-verbal cues during the interview process.

## Conclusion

We identified several barriers to implementing WHONET/BacLink in hospitals across Nepal. Limited user knowledge about the software's functionality, application, and benefits led to low confidence and motivation in adopting the system. Additionally, hospitals submitted Excel files for reporting to the GLASS Surveillance Network instead of the required SQLite format, highlighting an opportunity for a centralized coordination model where a designated coordinator could run BacLink for reporting while ensuring an efficient and sustainable data flow at individual sites without overwhelming end-users. Furthermore, the lack of integration between WHONET/BacLink and existing EMR systems resulted in redundant tasks for laboratory staff, with no direct financial or operational incentives.

From a broader policy and global perspective, similar challenges have been observed in other low-resource settings where CAPTURA project has been implemented, where fragmented digital infrastructures and limited technical capacity hinder the seamless adoption of laboratory surveillance tools. To address these challenges, we recommend an immediate and comprehensive assessment of Nepal's existing LIS infrastructure and digital ecosystem to identify barriers and opportunities for developing an integrated, scalable, and interoperable AMR surveillance system that aligns with global best practices.

## Supporting information

**S1 Table. Responses from the participants categorized as per the themes.**
(DOCX)

**S2 Table. Post-training assessment interview guideline.**
(DOCX)

**S3 Table. Summary table on implementation barriers and possible solutions.**
(DOCX)

**S4 Table. De-identified data supporting the findings of this study.**
(DOCX)

## Acknowledgments

Acknowledgement to all the CAPTURA sites stakeholders participating directly and indirectly through the hospital for this follow-up for effective capacity building activity. The views expressed in this publication are those of the authors and not necessarily those of the UK DHSC or its Management Agent, Mott MacDonald.

## Author contributions

**Conceptualization:** Florian Marks, Nimesh Poudyal.

**Data curation:** Reena Mukhiya, Smirti Kattel, Amit Bhandari, John Stelling.

**Formal analysis:** Sanju Maharjan, John Stelling.

**Funding acquisition:** Florian Marks, Nimesh Poudyal.

**Investigation:** Sanju Maharjan, Patrick Gallagher, Manish Gautam, Manisha Budhathoki, Ahmed Taha Aboushady, William R. MacWright, John Stelling, Nimesh Poudyal.

**Methodology:** Sanju Maharjan, Patrick Gallagher, Manish Gautam, Sanjay Gautam, Reena Mukhiya, Smirti Kattel, Ahmed Taha Aboushady, Raphaël M. Zellweger, William R. MacWright, John Stelling, Nimesh Poudyal.

**Project administration:** Sanju Maharjan, Manish Gautam, Manisha Budhathoki, Hea Sun Joh, Madan Kumar Upadhyaya, Runa Jha, Jyoti Acharya, Florian Marks, Nimesh Poudyal.

**Resources:** Manish Gautam, Hea Sun Joh, Ahmed Taha Aboushady, Raphaël M. Zellweger, Madan Kumar Upadhyaya, Runa Jha, Jyoti Acharya, William R. MacWright, Florian Marks, John Stelling, Nimesh Poudyal.

**Software:** John Stelling.

**Supervision:** Sanjay Gautam, Florian Marks, Nimesh Poudyal.

**Validation:** Patrick Gallagher, Sanjay Gautam, Florian Marks, John Stelling, Nimesh Poudyal.

**Writing – original draft:** Sanju Maharjan.

**Writing – review & editing:** Sanju Maharjan, Patrick Gallagher, Manish Gautam, Sanjay Gautam, Manisha Budhathoki, Reena Mukhiya, Smirti Kattel, Amit Bhandari, Hea Sun Joh, Ahmed Taha Aboushady, Raphaël M. Zellweger, Madan Kumar Upadhyaya, Runa Jha, Jyoti Acharya, William R. MacWright, Florian Marks, John Stelling, Nimesh Poudyal.

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
