## [Decision Letter · Decision Letter 0]

PONE-D-24-53694End users’ perception and challenges to adopt the WHONET/BacLink system in clinical microbiology laboratories in NepalPLOS ONE

Dear Dr. Gautam,

Thank you for submitting your manuscript to PLOS ONE. After careful consideration, we feel that it has merit but does not fully meet PLOS ONE’s publication criteria as it currently stands. Therefore, we invite you to submit a revised version of the manuscript that addresses the points raised during the review process.

We look forward to receiving your revised manuscript.

Kind regards,

Cornelius Cecil Dodoo, PhD

Academic Editor

PLOS ONE

2. Please amend either the title on the online submission form (via Edit Submission) or the title in the manuscript so that they are identical.

4. Please provide captions for Figure 1, Figure 2 in your manuscript.

6. In the ethics statement in the Methods, you have specified that verbal consent was obtained. Please provide additional details regarding how this consent was documented and witnessed, and state whether this was approved by the IRB.

7.  We note that the grant information you provided in the ‘Funding Information’ and ‘Financial Disclosure’ sections do not match.

8. Thank you for stating the following financial disclosure:

 [The "Capturing Data on Antimicrobial Resistance Patterns and Trends in Use in Regions of Asia (CAPTURA)," project at the International Vaccine Institute was funded by the Department of Health and Social Care’s (DHSC) Fleming Fund using UK aid.]. 

9. Thank you for stating the following in the Acknowledgments Section of your manuscript:

[Acknowledgement to all the CAPTURA sites stakeholder participating directly and indirectly through the hospital for this follow up of effective of capacity building activity. The "Capturing Data on Antimicrobial Resistance Patterns and Trends in Use in Regions of Asia (CAPTURA)," was funded by the Department of Health and Social Care’s (DHSC) Fleming Fund using UK aid. The views expressed in this publication are those of the authors and not necessarily those of the UK DHSC or its Management Agent, Mott MacDonald.]

[The "Capturing Data on Antimicrobial Resistance Patterns and Trends in Use in Regions of Asia (CAPTURA)," project at the International Vaccine Institute was funded by the Department of Health and Social Care’s (DHSC) Fleming Fund using UK aid.]. 

10. We note that you have indicated that there are restrictions to data sharing for this study. For studies involving human research participant data or other sensitive data, we encourage authors to share de-identified or anonymized data. However, when data cannot be publicly shared for ethical reasons, we allow authors to make their data sets available upon request. For information on unacceptable data access restrictions, please see http://journals.plos.org/plosone/s/data-availability#loc-unacceptable-data-access-restrictions.

11.  We notice that your supplementary tables are included in the manuscript file. Please remove them and upload them with the file type 'Supporting Information'. Please ensure that each Supporting Information file has a legend listed in the manuscript after the references list.

12. Please include captions for your Supporting Information files at the end of your manuscript, and update any in-text citations to match accordingly. Please see our Supporting Information guidelines for more information: http://journals.plos.org/plosone/s/supporting-information.

Reviewers' comments:

Reviewer's Responses to Questions

**Comments to the Author**

1. Is the manuscript technically sound, and do the data support the conclusions?

Reviewer #1: Partly

Reviewer #2: Yes

2. Has the statistical analysis been performed appropriately and rigorously? 

Reviewer #1: I Don't Know

Reviewer #2: Yes

3. Have the authors made all data underlying the findings in their manuscript fully available?

Reviewer #1: No

Reviewer #2: Yes

4. Is the manuscript presented in an intelligible fashion and written in standard English?

Reviewer #1: Yes

Reviewer #2: Yes

5. Review Comments to the Author

Reviewer #1: This manuscript, "End users’ perception and challenges to adopt the WHONET/BacLink system in clinical microbiology laboratories in Nepal," has potential for publication particularly in the context of global health, antimicrobial resistance (AMR) surveillance, and digital health interventions.

Although it presents a real-world evaluation of WHONET/BacLink and provides actionable insights into barriers to software adoption, however this manuscript has some key gaps that need to be addressed. Fo example,

• The title and abstract should be more explicit about the study's qualitative nature. Consider modifying the title to:

"Barriers and Perceptions of WHONET/BacLink Adoption in Nepal: A Qualitative Study of Clinical Microbiology Laboratories."

• The introduction should explicitly state the research question(s).

• The discussion section should have a more structured approach, aligning findings with solutions.

In methodology,

• the sample size (10 hospitals out of 19) is small, which limits generalizability. A discussion of this limitation and possible biases (e.g., response bias in phone interviews) is needed.

• The inductive coding approach used for qualitative analysis should be described more transparently (e.g., how themes were validated).

Data utilization & interoperability solutions

• Line 15-18: Despite given extensive training on uses of WHONET/BacLink, why do you think lack of knowledge is the barrier?

• The paper extensively discusses barriers but does not explore technical solutions to WHONET/BacLink's interoperability issues. Are there existing case studies or proposed solutions in other countries?

• The manuscript does not sufficiently compare hospitals that adopted the system vs. those that did not—are there contextual factors that influenced successful implementation?

• The study is valuable but lacks a broader policy or global perspective—how do these findings compare to other low-resource settings?

• Some findings (e.g., hospital-specific WHONET usage patterns) would be better presented in tables or figures for clarity.

Reviewer #2: The authors have put together a clear and well-written manuscript that adds to the narrative in the use of WHONET; an End users’ perception and challenges to adopting the WHONET/BacLink system in clinical microbiology laboratories in Nepal

find below a few suggestions.

The background in the abstract did not really capture the aim of the work/study if you could be modified to bring out the aim.

In the method section, was the translation done manually or was software used for the translation

Line 24 “IVI’ kindly gives the full meaning of this acronym as it is the first time appearing in the writeup

6. PLOS authors have the option to publish the peer review history of their article (what does this mean?). If published, this will include your full peer review and any attached files.

Reviewer #1: **Yes: **Md Hafizur Rahman

Reviewer #2: No

---

## [Author Response · Author response to Decision Letter 1]

28 May 2025

Dear reviewers,

As per the comments received, we have made following response:

1. The title and abstract should be more explicit about the study's qualitative nature. Consider modifying the title to:

"Barriers and Perceptions of WHONET/BacLink Adoption in Nepal: A Qualitative Study of Clinical Microbiology Laboratories."

Response: The title has been changed accordingly

2. The introduction should explicitly state the research question(s). Response: Page 3 (Line 20-24) has been added stating the research question

3. The discussion section should have a more structured approach, aligning findings with solutions. Response: Based on the feedback, the discussion section has been revised and findings based on possible contextual solutions have also been provided in each paragraph

4. In methodology,

• Ihe sample size (10 hospitals out of 19) is small, which limits generalizability. A discussion of this limitation and possible biases (e.g., response bias in phone interviews) is needed. Response: The point has been added in limitation. Page14(line 2-3) and line 11-15.

• The inductive coding approach used for qualitative analysis should be described more transparently (e.g., how these themes were validated). Response: To specify the process of validation. The description has been added. Page 4 (23-25) and Page 5 (Line 1)

5. Data utilization & interoperability solutions

- Line 15-18: Despite extensive training on uses of WHONET/BacLink, why do you think lack of knowledge is the barrier? Response: The sentences have been rephrased in page 10 Line (2-4).

- The paper extensively discusses barriers but does not explore technical solutions to WHONET/BacLink's interoperability issues. Are there existing case studies or proposed solutions in other countries? Response: Few solutions based on the findings and its analysis has been added in page 13 (line 13-16) and reference to other countries stating the similar problem has also been cited in page 12 (Line 24-25) and 13 (Line 1-5).

Since this study is a follow-up study to CAPTURA intervention, the key objective of the study was to identify the barriers and issues in implementing the WHONET BacLink software. The study guidelines were framed accordingly and we therefore had limitations to provide detailed solution specific discussion in this paper.

6. The manuscript does not sufficiently compare hospitals that adopted the system vs. those that did not—are there contextual factors that influenced successful implementation? Response: Thank you for your insightful suggestion but this was out of scope for this study, and we unfortunately did not compare hospitals adopting the system with those not adopting. This has been mentioned in limitations (page 14, line 5-7)

7. The study is valuable but lacks a broader policy or global perspective—how do these findings compare to other low-resource settings? – Response: Not many studies with supporting and contrasting findings to this study were identified but within the CAPTURA project sites, similar findings were observed, and it has been cited in page 12 (Line 24-25).

8. Some findings (e.g., hospital-specific WHONET usage patterns) would be better presented in tables or figures for clarity. Response: Thank you for your suggestion. As this study was entirely qualitative, we did not collect or analyze WHONET performance data. Instead, we focused on exploring users' experiences, motivations, and the barriers they faced, which could not be quantified. Additionally, we did not gather data on records uploaded to WHONET due to predefined ethical and data-sharing agreements between hospitals.

Update: 02 April 2025

As per the correspondence received, we have:

1. linked the ORCiD ID to the submission

2. Added deidentified data to suppporting information

Thank you

---

## [Editor Report · Decision Letter 1]

Barriers and perceptions of WHONET/BacLink adoption in Nepal: A qualitative study of clinical microbiology laboratories

PONE-D-24-53694R1

Dear Dr. Gautam,

We’re pleased to inform you that your manuscript has been judged scientifically suitable for publication and will be formally accepted for publication once it meets all outstanding technical requirements.

Kind regards,

Cornelius Cecil Dodoo, PhD

Academic Editor

PLOS ONE
---

## [Editor Report · Acceptance letter]

PONE-D-24-53694R1

PLOS ONE

Dear Dr. Gautam,

I'm pleased to inform you that your manuscript has been deemed suitable for publication in PLOS ONE. Congratulations! Your manuscript is now being handed over to our production team.

Kind regards,

on behalf of

Dr. Cornelius Cecil Dodoo

Academic Editor

PLOS ONE